# Generalized Linear Model Regression under Distance-to-set Penalties

**Jason Xu**
University of California, Los Angeles
jqxu@ucla.edu

**Eric C. Chi**
North Carolina State University
eric_chi@ncsu.edu

**Kenneth Lange**
University of California, Los Angeles
klange@ucla.edu

## Abstract

Estimation in generalized linear models (GLM) is complicated by the presence of constraints. One can handle constraints by maximizing a penalized log-likelihood. Penalties such as the lasso are effective in high dimensions, but often lead to unwanted shrinkage. This paper explores instead penalizing the squared distance to constraint sets. Distance penalties are more flexible than algebraic and regularization penalties, and avoid the drawback of shrinkage. To optimize distance penalized objectives, we make use of the majorization-minimization principle. Resulting algorithms constructed within this framework are amenable to acceleration and come with global convergence guarantees. Applications to shape constraints, sparse regression, and rank-restricted matrix regression on synthetic and real data showcase strong empirical performance, even under non-convex constraints.

## 1 Introduction and Background

In classical linear regression, the response variable $y$ follows a Gaussian distribution whose mean $\boldsymbol{x}^t\boldsymbol{\beta}$ depends linearly on a parameter vector $\boldsymbol{\beta}$ through a vector of predictors $\boldsymbol{x}$. Generalized linear models (GLMs) extend classical linear regression by allowing $y$ to follow any exponential family distribution, and the conditional mean of $y$ to be a nonlinear function $h(\boldsymbol{x}^t\boldsymbol{\beta})$ of $\boldsymbol{x}^t\boldsymbol{\beta}$ [24]. This encompasses a broad class of important models in statistics and machine learning. For instance, count data and binary classification come within the purview of generalized linear regression.

In many settings, it is desirable to impose constraints on the regression coefficients. Sparse regression is a prominent example. In high-dimensional problems where the number of predictors $n$ exceeds the number of cases $m$, inference is possible provided the regression function lies in a low-dimensional manifold [11]. In this case, the coefficient vector $\boldsymbol{\beta}$ is sparse, and just a few predictors explain the response $y$. The goals of sparse regression are to correctly identify the relevant predictors and to estimate their effect sizes. One approach, best subset regression, is known to be NP hard. Penalizing the likelihood by including an $\ell_0$ penalty $\|\boldsymbol{\beta}\|_0$ (the number of nonzero coefficients) is a possibility, but the resulting objective function is nonconvex and discontinuous. The convex relaxation of $\ell_0$ regression replaces $\|\boldsymbol{\beta}\|_0$ by the $\ell_1$ norm $\|\boldsymbol{\beta}\|_1$. This LASSO proxy for $\|\boldsymbol{\beta}\|_0$ restores convexity and continuity [31]. While LASSO regression has been a great success, it has the downside of simultaneously inducing both sparsity and parameter shrinkage. Unfortunately, shrinkage often has the undesirable side effect of including spurious predictors (false positives) with the true predictors.

Motivated by sparse regression, we now consider the alternative of penalizing the log-likelihood by the squared distance from the parameter vector $\boldsymbol{\beta}$ to the constraint set. If there are several constraints, then we add a distance penalty for each constraint set. Our approach is closely related to the proximal

distance algorithm [19, 20] and proximity function approaches to convex feasibility problems [5]. Neither of these prior algorithm classes explicitly considers generalized linear models. Beyond sparse regression, distance penalization applies to a wide class of statistically relevant constraint sets, including isotonic constraints and matrix rank constraints. To maximize distance penalized log-likelihoods, we advocate the majorization-minimization (MM) principle [2, 18, 19]. MM algorithms are increasingly popular in solving the large-scale optimization problems arising in statistics and machine learning [22]. Although distance penalization preserves convexity when it already exists, neither the objective function nor the constraints sets need be convex to carry out estimation. The capacity to project onto each constraint set is necessary. Fortunately, many projection operators are known. Even in the absence of convexity, we are able to prove that our algorithm converges to a stationary point. In the presence of convexity, the stationary points are global minima.

In subsequent sections, we begin by briefly reviewing GLM regression and shrinkage penalties. We then present our distance penalty method and a sample of statistically relevant problems that it can address. Next we lay out in detail our distance penalized GLM algorithm, discuss how it can be accelerated, summarize our convergence results, and compare its performance to that of competing methods on real and simulated data. We close with a summary and a discussion of future directions.

**GLMs and Exponential Families:** In linear regression, the vector of responses $\boldsymbol{y}$ is normally distributed with mean vector $\mathbb{E}(\boldsymbol{y}) = \boldsymbol{X}\boldsymbol{\beta}$ and covariance matrix $\mathbb{V}(\boldsymbol{y}) = \sigma^2 \boldsymbol{I}$. A GLM preserves the independence of the responses $y_i$ but assumes that they are generated from a shared exponential family distribution. The response $y_i$ is postulated to have mean $\mu_i(\boldsymbol{\beta}) = \mathbb{E}[y_i|\boldsymbol{\beta}] = h(\boldsymbol{x}_i^t\boldsymbol{\beta})$, where $\boldsymbol{x}_i$ is the $i$th row of a design matrix $\boldsymbol{X}$, and the inverse link function $h(s)$ is smooth and strictly increasing [24]. The functional inverse $h^{-1}(s)$ of $h(s)$ is called the link function. The likelihood of any exponential family can be written in the *canonical form*

$$p(y_i|\theta_i, \tau) = c_1(y_i, \tau) \exp\left\{\frac{y\theta_i - \psi(\theta_i)}{c_2(\tau)}\right\}. \tag{1}$$

Here $\tau$ is a fixed scale parameter, and the positive functions $c_1$ and $c_2$ are constant with respect to the natural parameter $\theta_i$. The function $\psi$ is smooth and convex; a brief calculation shows that $\mu_i = \psi'(\theta_i)$. The *canonical link* function $h^{-1}(s)$ is defined by the condition $h^{-1}(\mu_i) = \boldsymbol{x}_i^t\boldsymbol{\beta} = \theta_i$. In this case, $h(\theta_i) = \psi'(\theta_i)$, and the log-likelihood $\ln p(\boldsymbol{y}|\boldsymbol{\beta}, \boldsymbol{x}_j, \tau)$ is concave in $\boldsymbol{\beta}$. Because $c_1$ and $c_2$ are not functions of $\theta$, we may drop these terms and work with the log-likelihood up to proportionality. We denote this by $\mathcal{L}(\boldsymbol{\beta} \mid \boldsymbol{y}, \boldsymbol{X}) \propto \ln p(\boldsymbol{y}|\boldsymbol{\beta}, \boldsymbol{x}_j, \tau)$. The gradient and second differential of $\mathcal{L}(\boldsymbol{\beta} \mid \boldsymbol{y}, \boldsymbol{X})$ amount to

$$\nabla\mathcal{L} = \sum_{i=1}^{m}[y_i - \psi'(\boldsymbol{x}_i^t\boldsymbol{\beta})]\boldsymbol{x}_i \quad \text{and} \quad d^2\mathcal{L} = -\sum_{i=1}^{m}\psi''(\boldsymbol{x}_i^t\boldsymbol{\beta})\boldsymbol{x}_i\boldsymbol{x}_i^t. \tag{2}$$

As an example, when $\psi(\theta) = \theta^2/2$ and $c_2(\tau) = \tau^2$, the density (1) is the Gaussian likelihood, and GLM regression under the identity link coincides with standard linear regression. Choosing $\psi(\theta) = \ln[1 + \exp(\theta)]$ and $c_2(\tau) = 1$ corresponds to logistic regression under the canonical link $h^{-1}(s) = \ln\frac{s}{1-s}$ with inverse link $h(s) = \frac{e^s}{1+e^s}$. GLMs unify a range of regression settings, including Poisson, logistic, gamma, and multinomial regression.

**Shrinkage penalties:** The least absolute shrinkage and selection operator (LASSO) [12, 31] solves

$$\hat{\boldsymbol{\beta}} = \text{argmin}_{\boldsymbol{\beta}} \left[\lambda\|\boldsymbol{\beta}\|_1 - \frac{1}{m}\sum_{j=1}^{m}\mathcal{L}(\boldsymbol{\beta} \mid y_j, \boldsymbol{x}_j)\right], \tag{3}$$

where $\lambda > 0$ is a tuning constant that controls the strength of the $\ell_1$ penalty. The $\ell_1$ relaxation is a popular approach to promote a sparse solution, but there is no obvious map between $\lambda$ and the sparsity level $k$. In practice, a suitable value of $\lambda$ is found by cross-validation. Relying on global shrinkage towards zero, LASSO notoriously leads to biased estimates. This bias can be ameliorated by re-estimating under the model containing only the selected variables, known as the relaxed LASSO [25], but success of this two-stage procedure relies on correct support recovery in the first step. In many cases, LASSO shrinkage is known to introduce false positives [30], resulting in spurious covariates that cannot be corrected. To combat these shortcomings, one may replace the LASSO penalty by a non-convex penalty with milder effects on large coefficients. The smoothly clipped

absolute deviation (SCAD) penalty [10] and minimax concave penalty (MCP) [34] are even functions defined through their derivatives

$$q'_\gamma(\beta_i, \lambda) = \lambda \left[ \mathbf{1}_{\{|\beta_i| \le \lambda\}} + \frac{(\gamma\lambda - |\beta_i|)_+}{(\gamma - 1)\lambda} \mathbf{1}_{\{|\beta_i| > \lambda\}} \right] \quad \text{and} \quad q'_\gamma(\beta_i, \lambda) = \lambda \left( 1 - \frac{|\beta_i|}{\lambda\gamma} \right)_+$$

for $\beta_i > 0$. Both penalties reduce bias, interpolate between hard thresholding and LASSO shrinkage, and significantly outperform the LASSO in some settings, especially in problems with extreme sparsity. SCAD, MCP, as well as the relaxed lasso come with the disadvantage of requiring an extra tuning parameter $\gamma > 0$ to be selected.

## 2 Regression with distance-to-constraint set penalties

As an alternative to shrinkage, we consider penalizing the distance between the parameter vector $\boldsymbol{\beta}$ and constraints defined by sets $C_i$. Penalized estimation seeks the solution

$$\hat{\boldsymbol{\beta}} = \text{argmin}_{\boldsymbol{\beta}} \left[ \frac{1}{2} \sum_i v_i \text{dist}(\boldsymbol{\beta}, C_i)^2 - \frac{1}{m} \sum_{j=1}^m \mathcal{L}(\boldsymbol{\beta} \mid y_j, \boldsymbol{x}_j) \right] := \text{argmin}_{\boldsymbol{\beta}} f(\boldsymbol{\beta}), \qquad (4)$$

where the $v_i$ are weights on the distance penalty to constraint set $C_i$. The Euclidean distance can also be written as

$$\text{dist}(\boldsymbol{\beta}, C_i) = \|\boldsymbol{\beta} - P_{C_i}(\boldsymbol{\beta})\|_2,$$

where $P_{C_i}(\boldsymbol{\beta})$ denotes the projection of $\boldsymbol{\beta}$ onto $C_i$. The projection operator is uniquely defined when $C_i$ is closed and convex. If $C_i$ is merely closed, then $P_{C_i}(\boldsymbol{\beta})$ may be multi-valued for a few unusual external points $\boldsymbol{\beta}$. Notice the distance penalty $\text{dist}(\boldsymbol{\beta}, C_i)^2$ is 0 precisely when $\boldsymbol{\beta} \in C_i$. The solution (4) represents a tradeoff between maximizing the log-likelihood and satisfying the constraints. When each $C_i$ is convex, the objective function is convex as a whole. Sending all of the penalty constants $v_i$ to $\infty$ produces in the limit the constrained maximum likelihood estimate. This is the philosophy behind the proximal distance algorithm [19, 20]. In practice, it often suffices to find the solution (4) under fixed $v_i$ large. The reader may wonder why we employ squared distances rather than distances. The advantage is that squaring renders the penalties differentiable. Indeed, $\nabla \frac{1}{2}\text{dist}(\boldsymbol{x}, C_i)^2 = \boldsymbol{x} - P_{C_i}(\boldsymbol{x})$ whenever $P_{C_i}(\boldsymbol{x})$ is single valued. This is almost always the case. In contrast, $\text{dist}(\boldsymbol{x}, C_i)$ is typically nondifferentiable at boundary points of $C_i$ even when $C_i$ is convex. The following examples motivate distance penalization by considering constraint sets and their projections for several important models.

**Sparse regression:** Sparsity can be imposed directly through the constraint set $C_k = \{\boldsymbol{z} \in \mathbb{R}^n : \|\boldsymbol{z}\|_0 \le k\}$. Projecting a point $\boldsymbol{\beta}$ onto $C$ is trivially accomplished by setting all but the $k$ largest entries in magnitude of $\boldsymbol{\beta}$ equal to 0, the same operation behind iterative hard thresholding algorithms. Instead of solving the $\ell_1$-relaxation (3), our algorithm approximately solves the original $\ell_0$-constrained problem by repeatedly projecting onto the sparsity set $C_k$. Unlike LASSO regression, this strategy enables one to directly incorporate prior knowledge of the sparsity level $k$ in an interpretable manner. When no such information is available, $k$ can be selected by cross validation just as the LASSO tuning constant $\lambda$ is selected. Distance penalization escapes the NP hard dilemma of best subset regression at the cost of possible convergence to a local minimum.

**Shape and order constraints:** As an example of shape and order restrictions, consider isotonic regression [1]. For data $\boldsymbol{y} \in \mathbb{R}^n$, isotonic regression seeks to minimize $\frac{1}{2}\|\boldsymbol{y} - \boldsymbol{\beta}\|_2^2$ subject to the condition that the $\beta_i$ are non-decreasing. In this case, the relevant constraint set is the isotone convex cone $C = \{\boldsymbol{\beta} : \beta_1 \le \beta_2 \le \ldots \le \beta_n\}$. Projection onto $C$ is straightforward and efficiently accomplished using the pooled adjacent violators algorithm [1, 8]. More complicated order constraints can be imposed analogously: for instance, $\beta_i \le \beta_j$ might be required of all edges $i \to j$ in a directed graph model. Notably, isotonic linear regression applies to changepoint problems [32]; our approach allows isotonic constraints in GLM estimation. One noteworthy application is Poisson regression where the intensity parameter is assumed to be nondecreasing with time.

**Rank restriction:** Consider GLM regression where the predictors $\boldsymbol{X}_i$ and regression coefficients $\boldsymbol{B}$ are matrix-valued. To impose structure in high-dimensional settings, rank restriction serves as an

appropriate matrix counterpart to sparsity for vector parameters. Prior work suggests that imposing matrix sparsity is much less effective than restricting the rank of $\boldsymbol{B}$ in achieving model parsimony [37]. The matrix analog of the LASSO penalty is the nuclear norm penalty. The nuclear norm of a matrix $\boldsymbol{B}$ is defined as the sum of its singular values $\|\boldsymbol{B}\|_* = \sum_j \sigma_j(\boldsymbol{B}) = \mathrm{trace}(\sqrt{\boldsymbol{B}^*\boldsymbol{B}})$. Notice $\|\boldsymbol{B}\|_*$ is a convex relaxation of $\mathrm{rank}(\boldsymbol{B})$. Including a nuclear norm penalty entails shrinkage and induces low-rankness by proxy.

Distance penalization of rank involves projecting onto the set $C_r = \{\boldsymbol{Z} \in \mathbb{R}^{n \times n} : \mathrm{rank}(\boldsymbol{Z}) \leq r\}$ for a given rank $r$. Despite sacrificing convexity, distance penalization of rank is, in our view, both more natural and more effective than nuclear norm penalization. Avoiding shrinkage works to the advantage of distance penalization, which we will see empirically in Section 4. According to the Eckart-Young theorem, the projection of a matrix $\boldsymbol{B}$ onto $C_r$ is achieved by extracting the singular value decomposition of $\boldsymbol{B}$ and truncating all but the top $r$ singular values. Truncating the singular value decomposition is a standard numerical task best computed by Krylov subspace methods [14].

**Simple box constraints, hyperplanes, and balls:**  Many relevant set constraints reduce to closed convex sets with trivial projections. For instance, enforcing non-negative parameter values is accomplished by projecting onto the non-negative orthant. This is an example of a box constraint. Specifying linear equality and inequality constraints entails projecting onto a hyperplane or half-space, respectively. A Tikhonov or ridge penalty constraint $\|\boldsymbol{\beta}\|_2 \leq r$ requires spherical projection.

Finally, we stress that it is straightforward to consider combinations of the aforementioned constraints. Multiple norm penalties are already in common use. To encourage selection of correlated variables [38], the elastic net includes both $\ell_1$ and $\ell_2$ regularization terms. Further examples include matrix fitting subject to both sparse and low-rank matrix constraints [29] and LASSO regression subject to linear equality and inequality constraints [13]. In our setting the relative importance of different constraints can be controlled via the weights $v_i$.

# 3  Majorization-minimization

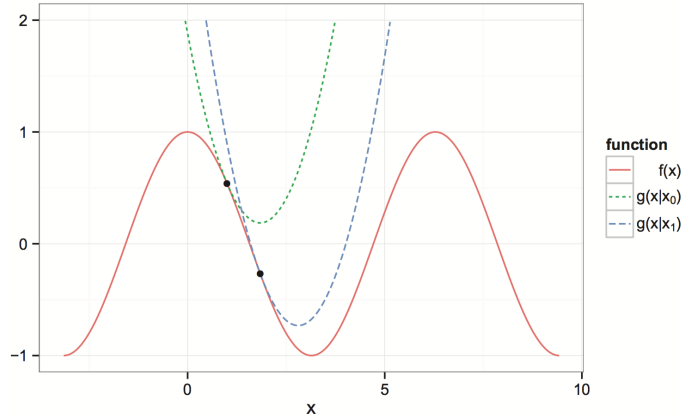

Figure 1: Illustrative example of two MM iterates with surrogates $g(x|x_k)$ majorizing $f(x) = \cos(x)$.

To solve the minimization problem (4), we exploit the principle of majorization-minimization. An MM algorithm successively minimizes a sequence of surrogate functions $g(\boldsymbol{\beta} \mid \boldsymbol{\beta}_k)$ majorizing the objective function $f(\boldsymbol{\beta})$ around the current iterate $\boldsymbol{\beta}_k$. See Figure 1. Forcing $g(\boldsymbol{\beta} \mid \boldsymbol{\beta}_k)$ downhill automatically drives $f(\boldsymbol{\beta})$ downhill as well [19, 22]. Every expectation-maximization (EM) algorithm [9] for maximum likelihood estimation is an MM algorithm. Majorization requires two conditions: tangency at the current iterate $g(\boldsymbol{\beta}_k \mid \boldsymbol{\beta}_k) = f(\boldsymbol{\beta}_k)$, and domination $g(\boldsymbol{\beta} \mid \boldsymbol{\beta}_k) \geq f(\boldsymbol{\beta})$ for all $\boldsymbol{\beta} \in \mathbb{R}^m$. The iterates of the MM algorithm are defined by

$$\boldsymbol{\beta}_{k+1} := \arg\min_{\boldsymbol{\beta}} g(\boldsymbol{\beta} \mid \boldsymbol{\beta}_k)$$

although all that is absolutely necessary is that $g(\boldsymbol{\beta}_{k+1} \mid \boldsymbol{\beta}_k) < g(\boldsymbol{\beta}_k \mid \boldsymbol{\beta}_k)$. Whenever this holds, the descent property

$$f(\boldsymbol{\beta}_{k+1}) \leq g(\boldsymbol{\beta}_{k+1} \mid \boldsymbol{\beta}_k) \leq g(\boldsymbol{\beta}_k \mid \boldsymbol{\beta}_k) = f(\boldsymbol{\beta}_k)$$

follows. This simple principle is widely applicable and converts many hard optimization problems (non-convex or non-smooth) into a sequence of simpler problems.

To majorize the objective (4), it suffices to majorize each distance penalty $\text{dist}(\boldsymbol{\beta}, C_i)^2$. The majorization $\text{dist}(\boldsymbol{\beta}, C_i)^2 \leq \|\boldsymbol{\beta} - P_{C_i}(\boldsymbol{\beta}_k)\|_2^2$ is an immediate consequence of the definitions of the set distance $\text{dist}(\boldsymbol{\beta}, C_i)^2$ and the projection operator $P_{C_i}(\boldsymbol{\beta})$ [8]. The surrogate function

$$g(\boldsymbol{\beta} \mid \boldsymbol{\beta}_k) \;\;=\;\; \frac{1}{2}\sum_i v_i\|\boldsymbol{\beta} - P_{C_i}(\boldsymbol{\beta}_k)\|_2^2 - \frac{1}{m}\sum_{j=1}^m \mathcal{L}(\boldsymbol{\beta} \mid y_j, \boldsymbol{x}_j).$$

has gradient

$$\nabla g(\boldsymbol{\beta} \mid \boldsymbol{\beta}_k) \;\;=\;\; \sum_i v_i[\boldsymbol{\beta} - P_{C_i}(\boldsymbol{\beta}_k)] - \frac{1}{m}\sum_{j=1}^m \nabla\mathcal{L}(\boldsymbol{\beta} \mid y_j, \boldsymbol{x}_j)$$

and second differential

$$d^2 g(\boldsymbol{\beta} \mid \boldsymbol{\beta}_k) \;\;=\;\; \Big(\sum_i v_i\Big)\boldsymbol{I}_n - \frac{1}{m}\sum_{j=1}^m d^2\mathcal{L}(\boldsymbol{\beta} \mid y_j, \boldsymbol{x}_j) := \boldsymbol{H}_k. \tag{5}$$

The score $\nabla\mathcal{L}(\boldsymbol{\beta} \mid y_j, \boldsymbol{x}_j)$ and information $-d^2\mathcal{L}(\boldsymbol{\beta} \mid y_j, \boldsymbol{x}_j)$ appear in equation (2). Note that for GLMs under canonical link, the observed and expected information matrices coincide, and their common value is thus positive semidefinite. Adding a multiple of the identity $\boldsymbol{I}_n$ to the information matrix is analogous to the Levenberg-Marquardt maneuver against ill-conditioning in ordinary regression [26]. Our algorithm therefore naturally benefits from this safeguard.

Since solving the stationarity equation $\nabla g(\boldsymbol{\beta} \mid \boldsymbol{\beta}_k) = \boldsymbol{0}$ is not analytically feasible in general, we employ one step of Newton's method in the form

$$\boldsymbol{\beta}_{k+1} = \boldsymbol{\beta}_k - \eta_k d^2 g(\boldsymbol{\beta}_k \mid \boldsymbol{\beta}_k)^{-1}\nabla f(\boldsymbol{\beta}_k),$$

where $\eta_k \in (0, 1]$ is a stepsize multiplier chosen via backtracking. Note here our application of the gradient identity $\nabla f(\boldsymbol{\beta}_k) = \nabla g(\boldsymbol{\beta}_k \mid \boldsymbol{\beta}_k)$, valid for all smooth surrogate functions. Because the Newton increment is a descent direction, some value of $\eta_k$ is bound to produce a decrease in the surrogate and therefore in the objective. The following theorem, proved in the Supplement, establishes global convergence of our algorithm under simple Armijo backtracking for choosing $\eta_k$:

**Theorem 3.1** *Consider the algorithm map*

$$\mathcal{M}(\boldsymbol{\beta}) = \boldsymbol{\beta} - \eta_{\boldsymbol{\beta}}\boldsymbol{H}(\boldsymbol{\beta})^{-1}\nabla f(\boldsymbol{\beta}),$$

*where the step size $\eta_{\boldsymbol{\beta}}$ has been selected by Armijo backtracking. Assume that $f(\boldsymbol{\beta})$ is coercive in the sense $\lim_{\|\boldsymbol{\beta}\|\to\infty} f(\boldsymbol{\beta}) = +\infty$. Then the limit points of the sequence $\boldsymbol{\beta}_{k+1} = \mathcal{M}(\boldsymbol{\beta}_k)$ are stationary points of $f(\boldsymbol{\beta})$. Moreover, the set of limit points is compact and connected.*

We remark that stationary points are necessarily global minimizers when $f(\boldsymbol{\beta})$ is convex. Furthermore, coercivity of $f(\boldsymbol{\beta})$ is a very mild assumption, and is satisfied whenever either the distance penalty or the negative log-likelihood is coercive. For instance, the negative log-likelihoods of the Poisson and Gaussian distributions are coercive functions. While this is not the case for the Bernoulli distribution, adding a small $\ell_2$ penalty $\omega\|\boldsymbol{\beta}\|_2^2$ restores coerciveness. Including such a penalty in logistic regression is a common remedy to the well-known problem of numerical instability in parameter estimates caused by a poorly conditioned design matrix $\boldsymbol{X}$ [27]. Since $\mathcal{L}(\boldsymbol{\beta})$ is concave in $\boldsymbol{\beta}$, the compactness of one or more of the constraint sets $C_i$ is another sufficient condition for coerciveness.

**Generalization to Bregman divergences:** Although we have focused on penalizing GLM likelihoods with Euclidean distance penalties, this approach holds more generally for objectives containing non-Euclidean measures of distance. As reviewed in the Supplement, the *Bregman divergence* $D_\phi(\boldsymbol{v}, \boldsymbol{u}) = \phi(\boldsymbol{v}) - \phi(\boldsymbol{u}) - d\phi(\boldsymbol{u})(\boldsymbol{v} - \boldsymbol{u})$ generated by a convex function $\phi(\boldsymbol{v})$ provides a general notion of directed distance [4]. The Bregman divergence associated with the choice $\phi(\boldsymbol{v}) = \frac{1}{2}\|\boldsymbol{v}\|_2^2$, for instance, is the squared Euclidean distance. One can rewrite the GLM penalized likelihood as a sum of multiple Bregman divergences

$$f(\boldsymbol{\beta}) = \sum_i v_i D_\phi\Big[\mathcal{P}_{C_i}^\phi(\boldsymbol{\beta}), \boldsymbol{\beta}\Big] + \sum_{j=1}^m w_j D_\zeta\Big[\boldsymbol{y}_j, \widetilde{h}_j(\boldsymbol{\beta})\Big]. \tag{6}$$

---

**Algorithm 1** MM algorithm to solve distance-penalized objective (4)

---

1: Initialize $k = 0$, starting point $\boldsymbol{\beta}_0$, initial step size $\alpha \in (0, 1)$, and halving parameter $\sigma \in (0, 1)$:
2: **repeat**
3:      $\nabla f_k \leftarrow \sum_i v_i [\boldsymbol{\beta} - P_{C_i}(\boldsymbol{\beta}_k)] - \frac{1}{m} \sum_{j=1}^m \nabla \mathcal{L}(\boldsymbol{\beta} \mid y_j, \boldsymbol{\beta}_j)$
4:      $\boldsymbol{H}_k \leftarrow \left( \sum_i v_i \right) \boldsymbol{I}_n - \frac{1}{m} \sum_{j=1}^m d^2 \mathcal{L}(\boldsymbol{\beta} \mid y_j, \boldsymbol{\beta}_j)$
5:      $\boldsymbol{v} \leftarrow -\boldsymbol{H}_k^{-1} \nabla f_k$
6:      $\eta \leftarrow 1$
7:      **while** $f(\boldsymbol{\beta}_k + \eta \boldsymbol{v}) > f(\boldsymbol{\beta}_k) + \alpha \eta \nabla f_k^t \boldsymbol{\beta}_k$ **do**
8:          $\eta \leftarrow \sigma \eta$
9:      **end while**
10:      $\boldsymbol{\beta}_{k+1} \leftarrow \boldsymbol{\beta}_k + \eta \boldsymbol{v}$
11:      $k \leftarrow k + 1$
12: **until** convergence

---

The first sum in equation (6) represents the distance penalty to the constraint sets $C_i$. The projection $\mathcal{P}_{C_i}^\phi(\boldsymbol{\beta})$ denotes the closest point to $\boldsymbol{\beta}$ in $C_i$ measured under $D_\phi$. The second sum generalizes the GLM log-likelihood term where $\widetilde{h}_j(\boldsymbol{\beta}) = h^{-1}(\boldsymbol{x}_j^t \boldsymbol{\beta})$. Every exponential family likelihood uniquely corresponds to a Bregman divergence $D_\zeta$ generated by the conjugate of its cumulant function $\zeta = \psi^*$ [28]. Hence, $-\mathcal{L}(\boldsymbol{\beta} \mid \boldsymbol{y}, \boldsymbol{X})$ is proportional to $\frac{1}{m} \sum_{j=1}^m D_\zeta \left[ \boldsymbol{y}_j, h^{-1}(\boldsymbol{x}_j^t \boldsymbol{\beta}) \right]$. The functional form (6) immediately broadens the class of objectives to include quasi-likelihoods and distances to constraint sets measured under a broad range of divergences. Objective functions of this form are closely related to proximity function minimization in the convex feasibility literature [5, 6, 7, 33]. The MM principle makes possible the extension of the projection algorithms of [7] to minimize this general objective.

Our MM algorithm for distance penalized GLM regression is summarized in Algorithm 1. Although for the sake of clarity the algorithm is written for vector-valued arguments, it holds more generally for matrix-variate regression. In this setting the regression coefficients $\boldsymbol{B}$ and predictors $\boldsymbol{X}_i$ are matrix valued, and response $y_j$ has mean $h[\text{trace}(\boldsymbol{X}_i^t \boldsymbol{B})] = h[\text{vec}(\boldsymbol{X}_i)^t \text{vec}(\boldsymbol{B})]$. Here the vec operator stacks the columns of its matrix argument. Thus, the algorithm immediately applies if we replace $\boldsymbol{B}$ by $\text{vec}(\boldsymbol{B})$ and $\boldsymbol{X}_1, \ldots, \boldsymbol{X}_m$ by $\boldsymbol{X} = [\text{vec}(\boldsymbol{X}_1), \ldots, \text{vec}(\boldsymbol{X}_m)]^t$. Projections requiring the matrix structure are performed by reshaping $\text{vec}(\boldsymbol{B})$ into matrix form. In contrast to shrinkage approaches, these maneuvers obviate the need for new algorithms in matrix regression [37].

**Acceleration:** Here we mention two modifications to the MM algorithm that translate to large practical differences in computational cost. Inverting the $n$-by-$n$ matrix $d^2 g(\boldsymbol{\beta}_k \mid \boldsymbol{\beta}_k)$ naively requires $\mathcal{O}(n^3)$ flops. When the number of cases $m \ll n$, invoking the Woodbury formula requires solving a substantially smaller $m \times m$ linear system at each iteration. This computational savings is crucial in the analysis of the EEG data of Section 4. The Woodbury formula says

$$(v\boldsymbol{I}_n + \boldsymbol{U}\boldsymbol{V})^{-1} = v^{-1}\boldsymbol{I}_n - v^{-2}\boldsymbol{U}\left(\boldsymbol{I}_m + v^{-1}\boldsymbol{V}\boldsymbol{U}\right)^{-1}\boldsymbol{V}$$

when $\boldsymbol{U}$ and $\boldsymbol{V}$ are $n \times m$ and $m \times n$ matrices, respectively. Inspection of equations (2) and (5) shows that $d^2 g(\boldsymbol{\beta}_k \mid \boldsymbol{\beta}_k)$ takes the required form. Under Woodbury's formula the dominant computation is the matrix-matrix product $\boldsymbol{V}\boldsymbol{U}$, which requires only $\mathcal{O}(nm^2)$ flops. The second modification to the MM algorithm is quasi-Newton acceleration. This technique exploits secant approximations derived from iterates of the algorithm map to approximate the differential of the map. As few as two secant approximations can lead to orders of magnitude reduction in the number of iterations until convergence. We refer the reader to [36] for a detailed description of quasi-Newton acceleration and a summary of its performance on various high-dimensional problems.

## 4 Results and performance

We first compare the performance of our distance penalization method to leading shrinkage methods in sparse regression. Our simulations involve a sparse length $n = 2000$ coefficient vector $\boldsymbol{\beta}$ with 10 nonzero entries. Nonzero coefficients have uniformly random effect sizes. The entries of the design matrix $\boldsymbol{X}$ are $N(0, 0.1)$ Gaussian random deviates. We then recover $\boldsymbol{\beta}$ from undersampled responses

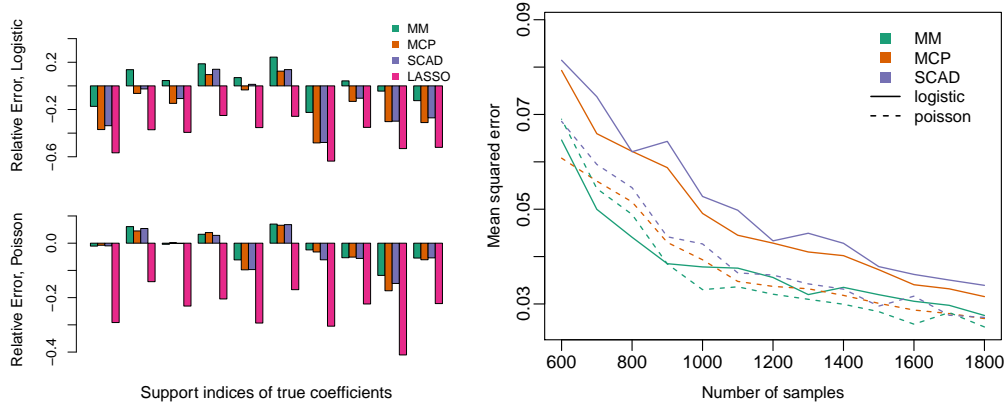

Figure 2: The left figure displays relative errors among nonzero predictors in underdetermined Poisson and logistic regression with $m = 1000$ cases. It is clear that LASSO suffers the most shrinkage and bias, while MM appears to outperform MCP and SCAD. The right figure displays MSE as a function of $m$, favoring MM most notably for logistic regression.

$y_j$ following Poisson and Bernoulli distributions with canonical links. Figure 2 compares solutions obtained using our distance penalties (MM) to those obtained under MCP, SCAD, and LASSO penalties. Relative errors (left) with $m = 1000$ cases clearly show that LASSO suffers the most shrinkage and bias; MM seems to outperform MCP and SCAD. For a more detailed comparison, the right side of the figure plots mean squared error (MSE) as a function of the number of cases averaged over 50 trials. All methods significantly outperform LASSO, which is omitted for scale, with MM achieving lower MSE than competitors, most noticeably in logistic regression. As suggested by an anonymous reviewer, similar results from additional experiments for Gaussian (linear) regression with comparison to relaxed lasso are included in the Supplement.

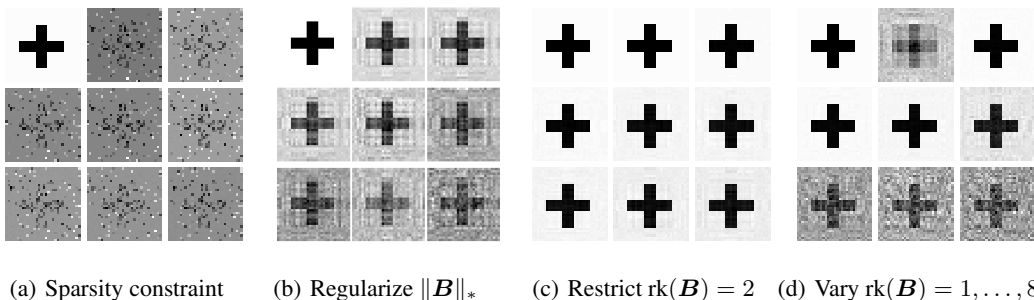

(a) Sparsity constraint     (b) Regularize $\|\boldsymbol{B}\|_*$     (c) Restrict $\mathrm{rk}(\boldsymbol{B}) = 2$    (d) Vary $\mathrm{rk}(\boldsymbol{B}) = 1, \ldots, 8$

Figure 3: True $\boldsymbol{B}_0$ in the top left of each set of 9 images has rank 2. The other 8 images in (a)—(c) display solutions as $\epsilon$ varies over the set $\{0, 0.1, \ldots, 0.7\}$. Figure (a) applies our MM algorithm with sparsity rather than rank constraints to illustrate how failing to account for matrix structure misses the true signal; Zhou and Li [37] report similar findings comparing spectral regularization to $\ell_1$ regularization. Figure (b) performs spectral shrinkage [37] and displays solutions under optimal $\lambda$ values via BIC, while (c) uses our MM algorithm restricting $\mathrm{rank}(\mathbf{B}) = 2$. Figure (d) fixes $\epsilon = 0.1$ and uses MM with $\mathrm{rank}(\mathbf{B}) \in \{1, \ldots, 8\}$ to illustrate robustness to rank over-specification.

For underdetermined matrix regression, we compare to the spectral regularization method developed by Zhou and Li [37]. We generate their cross-shaped $32 \times 32$ true signal $\boldsymbol{B}_0$ and in all trials sample $m = 300$ responses $y_i \sim N[\mathrm{tr}(\boldsymbol{X}_i^t, \boldsymbol{B}), \epsilon]$. Here the design tensor $\boldsymbol{X}$ is generated with standard normal entries. Figure 3 demonstrates that imposing sparsity alone fails to recover $\boldsymbol{Y}_0$ and that rank-set projections visibly outperform spectral norm shrinkage as $\epsilon$ varies. The rightmost panel also shows that our method is robust to over-specification of the rank of the true signal to an extent.

We consider two real datasets. We apply our method to count data of global temperature anomalies relative to the 1961-1990 average, collected by the Climate Research Unit [17]. We assume a non-

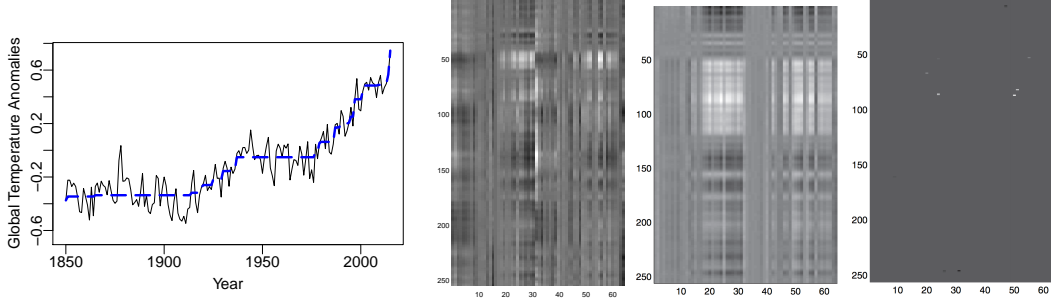

Figure 4: The leftmost plot shows our isotonic fit to temperature anomaly data [17]. The right figures display the estimated coefficient matrix $\boldsymbol{B}$ on EEG alcoholism data using distance penalization, nuclear norm shrinkage [37], and LASSO shrinkage, respectively.

decreasing solution, illustrating an instance of isotonic regression. The fitted solution displayed in Figure 4 has mean squared error 0.009, clearly obeys the isotonic constraint, and is consistent with that obtained on a previous version of the data [32]. We next focus on rank constrained matrix regression for electroencephalography (EEG) data, collected by [35] to study the association between alcoholism and voltage patterns over times and channels. The study consists of 77 individuals with alcoholism and 45 controls, providing 122 binary responses $y_i$ indicating whether subject $i$ has alcoholism. The EEG measurements are contained in $256 \times 64$ predictor matrices $\boldsymbol{X}_i$; the dimension $m$ is thus greater than $16,000$. Further details about the data appear in the Supplement.

Previous studies apply dimension reduction [21] and propose algorithms to seek the optimal rank 1 solution [16]. These methods could not handle the size of the original data directly, and the spectral shrinkage approach proposed in [37] is the first to consider the full EEG data. Figure 4 shows that our regression solution is qualitatively similar to that obtained under nuclear norm penalization [37], revealing similar time-varying patterns among channels 20-30 and 50-60. In contrast, ignoring matrix structure and penalizing the $\ell_1$ norm of $\boldsymbol{B}$ yields no useful information, consistent with findings in [37]. However, our distance penalization approach achieves a lower misclassification error of $0.1475$. The lowest misclassification rate reported in previous analyses is $0.139$ by [16]. As their approach is strictly more restrictive than ours in seeking a rank 1 solution, we agree with [37] in concluding that the lower misclassification error can be largely attributed to benefits from data preprocessing and dimension reduction. While not visually distinguishable, we also note that shrinking the eigenvalues via nuclear norm penalization [37] fails to produce a low-rank solution on this dataset.

We omit detailed timing comparisons throughout since the various methods were run across platforms and depend heavily on implementation. We note that MCP regression relies on the MM principle, and the LQA and LLA algorithms used to fit models with SCAD penalties are also instances of MM algorithms [11]. Almost all MM algorithms share an overall linear rate of convergence. While these require several seconds of compute time on a standard laptop machine, coordinate-descent implementations of LASSO outstrip our algorithm in terms of computational speed. Our MM algorithm required 31 seconds to converge on the EEG data, the largest example we considered.

## 5    Discussion

GLM regression is one of the most widely employed tools in statistics and machine learning. Imposing constraints upon the solution is integral to parameter estimation in many settings. This paper considers GLM regression under distance-to-set penalties when seeking a constrained solution. Such penalties allow a flexible range of constraints, and are competitive with standard shrinkage methods for sparse and low-rank regression in high dimensions. The MM principle yields a reliable solution method with theoretical guarantees and strong empirical results over a number of practical examples. These examples emphasize promising performance under non-convex constraints, and demonstrate how distance penalization avoids the disadvantages of shrinkage approaches.

Several avenues for future work may be pursued. The primary computational bottleneck we face is matrix inversion, which limits the algorithm when faced with extremely large and high-dimensional

datasets. Further improvements may be possible using modifications of the algorithm tailored to specific problems, such as coordinate or block descent variants. Since the linear systems encountered in our parameter updates are well conditioned, a conjugate gradient algorithm may be preferable to direct methods of solution in such cases. The updates within our algorithm can be recast as weighted least squares minimization, and a re-examination of this classical problem may suggest even better iterative solvers. As the methods apply to a generalized objective comprised of multiple Bregman divergences, it will be fruitful to study penalties under alternate measures of distance, and settings beyond GLM regression such as quasi-likelihood estimation.

While our experiments primarily compare against shrinkage approaches, an anonymous referee points us to recent work revisiting best subset selection using modern advances in mixed integer optimization [3]. These exciting developments make best subset regression possible for much larger problems than previously thought possible. As [3] focus on the linear case, it is of interest to consider how ideas in this paper may offer extensions to GLMs, and to compare the performance of such generalizations. Best subsets constitutes a gold standard for sparse estimation in the noiseless setting; whether it outperforms shrinkage methods seems to depend on the noise level and is a topic of much recent discussion [15, 23]. Finally, these studies as well as our present paper focus on estimation, and it will be fruitful to examine variable selection properties in future work. Recent work evidences an inevitable trade-off between false and true positives under LASSO shrinkage in the linear sparsity regime [30]. The authors demonstrate that this need not be the case with $\ell_0$ methods, remarking that computationally efficient methods which also enjoy good model performance would be highly desirable as $\ell_0$ and $\ell_1$ approaches possess one property but not the other [30]. Our results suggest that distance penalties, together with the MM principle, seem to enjoy benefits from both worlds on a number of statistical tasks.

**Acknowledgements:** We would like to thank Hua Zhou for helpful discussions about matrix regression and the EEG data. JX was supported by NSF MSPRF #1606177.

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
