[Supplementary Material]

# Supplemental Material: Generalized Linear Model Regression under Distance-to-set Penalties

**Jason Xu**
University of California, Los Angeles
jqxu@ucla.edu

**Eric C. Chi**
North Carolina State University
eric_chi@ncsu.edu

**Kenneth Lange**
University of California, Los Angeles
klange@ucla.edu

## 1   Proof of Convergence

We repeat the statement of Theorem 3.1 below:

**Theorem 1.1.** *Consider the algorithm map*

$$\mathcal{M}(\boldsymbol{\beta}) = \boldsymbol{\beta} - \eta_{\boldsymbol{\beta}} \boldsymbol{H}(\boldsymbol{\beta})^{-1} \nabla f(\boldsymbol{\beta}),$$

*where the step size $\eta_{\boldsymbol{\beta}}$ has been selected by Armijo backtracking. Assume that $f$ is coercive:* $\lim_{\|\boldsymbol{\beta}\| \to \infty} f(\boldsymbol{\beta}) = +\infty$. *Then the limit points of the sequence $\boldsymbol{\beta}_{k+1} = \mathcal{M}(\boldsymbol{\beta}_k)$ are stationary points of $f(\boldsymbol{\beta})$. Moreover, this set of limit points is compact and connected.*

Our algorithm selects the step-size $\eta$ according to the Armijo condition: suppose $\boldsymbol{v}$ is a descent direction at $\boldsymbol{\beta}$ in the sense that $df(\boldsymbol{\beta})\boldsymbol{v} < 0$. The Armijo condition chooses a step size $\eta$ such that

$$f(\boldsymbol{\beta} + t\boldsymbol{v}) \le f(\boldsymbol{\beta}) + \alpha\eta df(\boldsymbol{\beta})\boldsymbol{v},$$

for a constant $\alpha \in (0,1)$. Before proving the statement, the next lemma follows an argument in Chapter 12 of [3] to show that step-halving under the Armijo condition requires finitely many steps. We may then apply a similar argument used in [6].

**Lemma 1.2.** *Given $\alpha \in (0,1)$ and $\sigma \in (0,1)$, there exists an integer $s \ge 0$ such that*

$$f(\boldsymbol{\beta} + \sigma^s \boldsymbol{v}) \le f(\boldsymbol{\beta}) + \alpha\sigma^s df(\boldsymbol{\beta})\boldsymbol{v},$$

*where $\boldsymbol{v} = -\boldsymbol{H}(\boldsymbol{\beta})^{-1}\nabla f(\boldsymbol{\beta})$.*

*Proof.* Since $f$ is coercive by assumption, its sublevel sets are compact. Namely, the set $\mathcal{S}_f(\boldsymbol{\beta}_0) \equiv \{\boldsymbol{\beta} : f(\boldsymbol{\beta}) \le f(\boldsymbol{\beta}_0)\}$ is compact. Smoothness of the GLM likelihood and squared distance penalty ensure continuity of $\nabla f(\boldsymbol{\beta})$ and $\boldsymbol{H}(\boldsymbol{\beta})$. Together with coercivity, this implies that there exist positive constants $a$ and $b$, such that

$$\|\boldsymbol{H}(\boldsymbol{\beta})^{-1}\| \le a; \quad \|\boldsymbol{H}(\boldsymbol{\beta})\| \le b; \quad \|d^2\mathcal{L}(x)\| \le c$$

for all $\boldsymbol{\beta} \in \mathcal{S}_f(\boldsymbol{\beta}_0)$. Together with the fact that the Euclidean distance to a closed set dist$(\boldsymbol{\beta}, C)$ is a Lipschitz function with Lipschitz constant 1, we produce the inequality

$$f(\boldsymbol{\beta} + \eta\boldsymbol{v}) \le f(\boldsymbol{\beta}) + \eta df(\boldsymbol{\beta})\boldsymbol{v} + \frac{1}{2}\eta^2 L\|\boldsymbol{v}\|^2 \tag{1}$$

where $L = 1 + c$. The squared term appearing at the end of (1) can be bounded by

$$\|\boldsymbol{v}\|^2 = \|\boldsymbol{H}(\boldsymbol{\beta})^{-1}\nabla f(\boldsymbol{\beta})\|^2 \le a^2\|\nabla f(\boldsymbol{\beta})\|^2.$$

We next identify a bound for $\|\nabla f(\boldsymbol{\beta})\|^2$:

$$
\begin{aligned}
\|\nabla f(\boldsymbol{\beta})\|^2 &= \|\boldsymbol{H}(\boldsymbol{\beta})^{1/2}\boldsymbol{H}(\boldsymbol{\beta})^{-1/2}\nabla f(\boldsymbol{\beta})\|^2 \\
&\leq \|\boldsymbol{H}(\boldsymbol{\beta})^{1/2}\|^2\|\boldsymbol{H}(\boldsymbol{\beta})^{-1/2}\nabla f(\boldsymbol{\beta})\|^2 \\
&\leq bdf(\boldsymbol{\beta})\boldsymbol{H}(\boldsymbol{\beta})^{-1}\nabla f(\boldsymbol{\beta}) \\
&= -bdf(\boldsymbol{\beta})\boldsymbol{v}.
\end{aligned}
\tag{2}
$$

Combining inequalities (1) and (2) yields

$$
f(\boldsymbol{\beta} + \eta\boldsymbol{v}) \leq f(\boldsymbol{\beta}) + \eta\left(1 - \frac{a^2 bL}{2}t\right)df(\boldsymbol{\beta})\boldsymbol{v}.
$$

Thus, the Armijo condition is guaranteed to be satisfied as soon as $s \geq s^\star$, where

$$
s^\star = \frac{1}{\ln\sigma}\ln\left(\frac{2(1-\alpha)}{a^2 bL}\right).
$$

Of course, in practice a much lower value of $s$ may suffice.

$\square$

We are now ready to prove the original theorem: note the argument applies whenever $C$ is convex.

*Proof.* Consider the iterates of the algorithm $\boldsymbol{\beta}_{k+1} = \mathcal{M}(\boldsymbol{\beta}_k) = \boldsymbol{\beta}_k + \sigma^{s_k}\boldsymbol{v}_k$. Since $f(\boldsymbol{\beta})$ is continuous, $f$ attains its infimum over $\mathcal{S}_f(\boldsymbol{\beta}_0)$, and therefore the monotonically decreasing sequence $f(\boldsymbol{\beta}_k)$ is bounded below. This implies that $f(\boldsymbol{\beta}_k) - f(\boldsymbol{\beta}_{k+1})$ converges to 0. Let $s_k$ denote the number of backtracking steps taken at the $k$th iteration under the Armijo stopping rule. By Lemma 1.2, $s_k$ is finite, and thus

$$
\begin{aligned}
f(\boldsymbol{\beta}_k) - f(\boldsymbol{\beta}_{k+1}) &\geq -\alpha\sigma^{s_k}df(\boldsymbol{\beta}_k)\boldsymbol{v}_k \\
&= \alpha\sigma^{s_k}df(\boldsymbol{\beta}_k)\boldsymbol{H}(\boldsymbol{\beta}_k)^{-1}\nabla f(\boldsymbol{\beta}) \\
&\geq \frac{\alpha\sigma^{s_k}}{\beta}\|\nabla f(\boldsymbol{\beta}_k)\|^2 \\
&\geq \frac{\alpha\sigma^{s^\star+1}}{\beta}\|\nabla f(\boldsymbol{\beta}_k)\|^2.
\end{aligned}
$$

This inequality implies that $\|\nabla f(\boldsymbol{\beta}_k)\|$ converges to 0, and therefore all the limit points of the sequence $\boldsymbol{\beta}_k$ are stationary points of $f(\boldsymbol{\beta})$. Further, taking norms of the update yields the inequality

$$
\begin{aligned}
\|\boldsymbol{\beta}_{k+1} - \boldsymbol{\beta}_k\| &= \sigma^{s_k}\|\boldsymbol{H}(\boldsymbol{\beta}_k)^{-1}\nabla f(\boldsymbol{\beta}_k)\| \\
&\leq \sigma^{s_k}a\|\nabla f(\boldsymbol{\beta}_k)\|.
\end{aligned}
$$

Thus, the iterates $\boldsymbol{\beta}_k$ are a bounded sequence such that $\|\boldsymbol{\beta}_{k+1} - \boldsymbol{\beta}_k\|$ tends to 0, allowing us to conclude that the limit points form a compact and connected set by Propositions 12.4.2 and 12.4.3 in [3]. $\square$

## 2 Bregman Divergences

Let $\phi : \Omega \mapsto \mathbb{R}$ be a strictly convex function defined on a convex domain $\Omega \subset \mathbb{R}^n$ differentiable on the interior of $\Omega$. The Bregman divergence [1] between $\boldsymbol{u}$ and $\boldsymbol{v}$ with respect to $\phi$ is defined as

$$
D_\phi(\boldsymbol{v}, \boldsymbol{u}) = \phi(\boldsymbol{v}) - \phi(\boldsymbol{u}) - d\phi(\boldsymbol{u})(\boldsymbol{v} - \boldsymbol{u}).
\tag{3}
$$

Note that the Bregman divergence (3) is a convex function of its first argument $\boldsymbol{v}$, and measures the distance between $\boldsymbol{v}$ and a first order Taylor expansion of $\phi$ about $\boldsymbol{u}$ evaluated at $\boldsymbol{v}$. While the Bregman divergence is not a metric as it is not symmetric in general, it provides a natural notion of directed distance. It is non-negative for all $\boldsymbol{u}, \boldsymbol{v}$ and equal to zero if and only if $\boldsymbol{v} = \boldsymbol{u}$. Instances of Bregman divergences abound in statistics and machine learning, many useful measures of closeness.

Recall exponential family distributions takes the canonical form

$$p(y|\theta, \tau) = C_1(y, \tau) \exp\left\{ \frac{y\theta - \psi(\theta)}{C_2(\tau)} \right\}.$$

Each distribution belonging to an exponential family shares a close relationship to a Bregman divergence, and we may explicitly relate GLMs as a special case using this connection. Specifically, the conjugate of its cumulant function $\psi$, which we denote $\zeta$, uniquely generates a Bregman divergence $D_\zeta$ that represents the exponential family likelihood up to proportionality [5]. With $g$ denoting the link function, the negative log-likelihood of $y$ can be written as its Bregman divergence to the mean:

$$-\ln p(y|\theta, \tau) = D_\zeta \left( y, g^{-1}(\theta) \right) + C(y, \tau).$$

As an example, the cumulant function in the Poisson likelihood is $\psi(x) = e^x$, whose conjugate $\zeta(x) = x \ln x - x$ produces the relative entropy

$$D_\zeta(p, q) = p \ln(p/q) - p + q.$$

Similarly, recall that the Bernoulli likelihood in logistic regression has cumulant function $\psi(x) = \ln(1 + \exp(x))$. Its conjugate is given by $\zeta(x) = x \ln x + (1 - x) \ln(1 - x)$, and generates

$$D_\zeta(p, q) = p \ln \frac{p}{q} + (1 - p) \ln \frac{1 - p}{1 - q}.$$

This relationship implies that maximizing the likelihood in an exponential family is equivalent to minimizing a corresponding Bregman divergence between the data $\boldsymbol{y}$ and the regression coefficients $\boldsymbol{\beta}$. Notice this is a different statement than the well-known equivalence between maximizing the likelihood and minimizing the Kullback-Leibler divergence between the empirical and parametrized distributions. The gradients of the Bregman projection take the form

$$\nabla D_\phi \left( \mathcal{P}_{C_i}^\phi(\boldsymbol{\beta}_k), \boldsymbol{\beta} \right) = d^2\phi(\boldsymbol{\beta}) \left( \boldsymbol{\beta} - \mathcal{P}_{C_i}^\phi(\boldsymbol{\beta}_k) \right).$$

Further, the notion of Bregman divergence naturally applies to matrices:

$$D_\phi(\boldsymbol{V}, \boldsymbol{U}) = \phi(\boldsymbol{V}) - \phi(\boldsymbol{U}) - \langle \nabla\phi(\boldsymbol{U}), \boldsymbol{V} - \boldsymbol{U} \rangle$$

where $\langle \boldsymbol{V}, \boldsymbol{U} \rangle = \mathrm{Tr}(\boldsymbol{V}\boldsymbol{U}^T)$ denotes the inner product. For instance, the squared Frobenius distance between $\boldsymbol{V}, \boldsymbol{U}$ is generated by the choice of $\phi(\boldsymbol{V}) = \frac{1}{2}\|\boldsymbol{V}\|_F^2$. The MM algorithm therefore applies analogously to objective functions consisting of multiple Bregman divergences.

## 3   EEG dataset

The dataset we consider using rank restricted matrix regression seeks to study the association between alcoholism and the voltage patterns over times and channels from EEG data. The data are collected by [7], who provide further details of the experiment, and measures subjects over 120 trials. The study consists of 77 individuals with alcoholism and 45 controls. For each subject, 64 channels of electrodes were placed across the scalp, and voltages are recorded at 256 time points sampled at 256 Hz over one second. This is repeated over 120 trials with three different stimuli. Following the practice of previous studies of the data by [4, 2, 8], we consider covariates $\mathbf{X}$ representing the average over all trials of voltages recorded from each electrode. Other than averaging over trials, no data preprocessing is applied. $\mathbf{X}$ is thus a $256 \times 64$ matrix whose $ij$th entries represent the voltage at time $i$ in channel or electrode $j$, averaged over the 120 trials. The binary responses $y_i$ indicate whether subject $i$ has alcoholism.

As mentioned in the main text, the study by [4] focuses on reduction of the data via dimension folding, and the matrix-variate logistic regression algorithm proposed by [2] is also applied to preprocessed data using a generic dimension reduction technique. The nuclear norm shrinkage proposed by [8] is the first to consider matrix regression on the full, unprocessed data (apart from averaging over the 120 trials). The authors [8] point out that previous methods nonetheless attain better classification rates, likely due to the fact that preprocessing and tuning were chosen to optimize predictive accuracy. Indeed, the lowest misclassification rate reported in previous analyses is $0.139$ by [2], yet the authors show that their method is equivalent to seeking the best rank 1 approximation to the true coefficient matrix in terms of Kullback-Leibler divergence. Since this approach is strictly more restrictive than ours, which attains an error of $0.1475$, we agree with [8] in concluding that the lower misclassification error achieved by previous studies can be largely attributed to benefiting from removal of noise via data preprocessing and dimension reduction.

# 4 Additional comparisons in Gaussian regression

We consider an analogous simulation study including a comparison to the two-stage relaxed LASSO procedure, implemented in R package `relaxo`. The author's implementation is limited to the Gaussian case, and we consider linear regression with dimension $n = 2000$ as the number of samples $m$ varies, with $k = 12$ nonzero true coefficients. We consider a reduced experiment due to runtime considerations of `relaxo`, repeating only over 20 trials and varying $m$ by increments of 200. Though timing is heavily dependent on implementations, the average total runtimes (across all values of $m$) of the experiment across trials for MM, MCP, SCAD, and relaxed lasso are $96.8, 137.5, 107.3, 4876.6$ seconds, respectively. We see that relaxed LASSO is effective toward removing the bias induced by standard LASSO, and overall results are similar to those included in the main text.

Figure 1: Median MSE over 20 trials as a function of the number of samples $m$ in linear regression under our MM approach, the two-stage relaxed LASSO procedure, SCAD and MCP.