[Reviews · NeurIPS 2017]

Reviewer 1



The paper proposes a new approach for learning a generalized linear model that penalizes the distance between the weight parameters and a set. A majorization-minimization algorithm is presented for learning the model with the proposed penalty. The proposed approach of penalizing distance-to-set is potentially significant in that it provides a framework that encompasses many of the popular penalized (generalized) regression methods such as sparse regression, regression with rank restriction etc. The optimization algorithm generalizes well to other losses beyond the generalized linear models. The proposed approach could be of interest to many researchers in the community. The paper is well-written. The paper provides evidence that the proposed approach works well in the experiments section. The learning algorithm is reasonably fast for analysis of real-world datasets. Minor comments: It would be good to provide details of the temperature anomaly count data and mean squared errors on this dataset.

Reviewer 2



I found much to like about this paper. It was very clear and well-written, and the proposed method is elegant, intuitive, novel (to the best of my knowledge), seemingly well motivated, and widely applicable. In addition, the main ideas in the paper are well-supported by appropriate numerical and real data examples. I would like to see the authors compare their method to two other methods that I would see as major competitors: first, the relaxed LASSO (Meinshausen 2007) wherein we fit the lasso and then do an unpenalized fit using only the variables that the lasso has selected. As I understand the field, this is the most popular method and the one that Hastie, Tibshirani, and Wainwright (2015) recommend for users who want to avoid the bias of the lasso (but note that the shrinkage “bias” is sometimes desirable to counteract the selection bias or “winner’s curse” that the selected coefficients may suffer from). Second, best-subsets itself is now feasible using modern mixed-integer optimization methods (Bertsimas, Kind, and Mazumder, 2016) (if one of these competitors outperforms your method, it would not make me reconsider my recommendation to accept, since the method you propose applies to a much more general class of problems). Bertsimas, Dimitris, Angela King, and Rahul Mazumder. "Best subset selection via a modern optimization lens." The Annals of Statistics 44.2 (2016): 813-852. Hastie, Trevor, Robert Tibshirani, and Martin Wainwright. Statistical learning with sparsity: the lasso and generalizations. CRC press, 2015. Meinshausen, Nicolai. "Relaxed lasso." Computational Statistics & Data Analysis 52.1 (2007): 374-393.